# A Novel 3D Helical Microelectrode Array for In Vitro Extracellular Action Potential Recording

**DOI:** 10.3390/mi13101692

**Published:** 2022-10-08

**Authors:** Negar Geramifard, Jennifer Lawson, Stuart F. Cogan, Bryan James Black

**Affiliations:** 1Department of Bioengineering, Erik Jonsson School of Engineering and Computer Science, University of Texas at Dallas, Richardson, TX 75080, USA; 2Biomedical Engineering Department, Francis College of Engineering, University of Massachusetts Lowell, Lowell, MA 01854, USA

**Keywords:** 3D microelectrode, microelectrode arrays, iPSC sensory neurons, 3D cell culture

## Abstract

Recent advances in cell and tissue engineering have enabled long-term three-dimensional (3D) in vitro cultures of human-derived neuronal tissues. Analogous two-dimensional (2D) tissue cultures have been used for decades in combination with substrate integrated microelectrode arrays (MEA) for pharmacological and toxicological assessments. While the phenotypic and cytoarchitectural arguments for 3D culture are clear, 3D MEA technologies are presently inadequate. This is mostly due to the technical challenge of creating vertical electrical conduction paths (or ‘traces’) using standardized biocompatible materials and fabrication techniques. Here, we have circumvented that challenge by designing and fabricating a novel helical 3D MEA comprised of polyimide, amorphous silicon carbide (a-SiC), gold/titanium, and sputtered iridium oxide films (SIROF). Electrochemical impedance spectroscopy (EIS) and cyclic voltammetry (CV) testing confirmed fully-fabricated MEAs should be capable of recording extracellular action potentials (EAPs) with high signal-to-noise ratios (SNR). We then seeded induced pluripotent stems cell (iPSC) sensory neurons (SNs) in a 3D collagen-based hydrogel integrated with the helical MEAs and recorded EAPs for up to 28 days in vitro from across the MEA volume. Importantly, this highly adaptable design does not intrinsically limit cell/tissue type, channel count, height, or total volume.

## 1. Introduction

Microelectrode arrays (MEAs) enable the recording of extracellular action potentials (EAPs) from electrogenic cells (e.g., neurons or cardiomyocytes). In vitro, substrate-integrated MEAs have been widely adopted in academic/translational research for phenotypic and preclinical pharmacological screening of candidate drugs [1,2] and potential toxicants [3,4,5,6] due to their advantages as compared to traditional patch clamp recordings [7,8] and in vivo animal behavioral models [9]. In the past, in vitro cell culture-based models have been ‘seeded’ on 2D substrates, where cells develop focal adhesions on a single (often highly rigid) planar substrate and have limited cell-to-cell and cell-to-ligand interactions [10,11,12]. More recently, academic and industry researchers have demonstrated viable self-assembled and bioprinted [13,14] 3D human tissue models. These may be especially relevant for studying the function of, or assaying against, disease states in tissues with complex native immune-architectures, such as those of the central and peripheral nervous system [13,14,15]. To date, and to our knowledge, for EAP recordings, there have been four general methodologies employed. (1) Grow or place and wrap a 3D organoid on/in a 2D, substrate integrated MEA [16,17]. (2) Generate a matured organoid model and then pierce the organoid using a stiff MEA [18]. (3) Generate a matured organoid and record from surface neurons using patch clamp techniques [19]. (4) Grow and mature an organoid in and around an existing 3D MEA [20] that integrates directly into the ECM [21]. Presumably, the fourth option would allow for the most intimate interface with multi-site MEAs without damaging a pre-existing tissue model/organoid. However, there are technical constraints to applying common microfabrication techniques to 3D MEAs. For example, thin film lithography approaches utilizing biocompatible polymers typically develop film layers on the micrometer scale. This approach is not conducive to vertical construction of millimeter-scaled recording devices; nor is it practical to adopt top-down etching techniques as this constrains the placement of electrode sites along the length of any given vertical ‘shank’, essentially rendering a 2D recording plane, though elevated. Soscia et al. overcame this challenge by fabricating hinged thin-film 2D shanks and then mechanically actuating the shanks into standing (3D) positions [20]. However, this approach, although functionally successful, has issues regarding the precision and reproducibility of mechanical actuation and the conductive integrity of the ‘hinge’. Here, we report the fabrication and testing of a novel 3D helical multi-site MEA. This MEA design enables true 3D access to tissue or organoids which may be organized on the MEA substrate and does not require sharp bends of the conductive traces. It is based on standard thin-film lithography techniques and incorporates standard biocompatible conducting and insulating materials (i.e., [22] and [23], respectively). Mono-cultures of induced pluripotent stem cell (iPSC) sensory neurons (SNs) were embedded in a thermo-polymerized hydrogel, integrated with the 1 mm high helical MEA, and EAP recordings were collected up to 28 days in vitro (DIV) with acceptable signal-to-noise ratios (SNR). Pharmacology and immunocytochemistry confirmed iPSC SN viability, biological sourcing of recorded signals, and persistent adjacency to electrodes throughout the recording volume. This preliminary work demonstrates the viability of 3D helical MEAs and offers a novel path to higher channel count for organoid or tissue mimetic recordings.

## 2. Materials and Methods

### 2.1. MEA Fabrication

In order to fabricate 3D helical MEAs, planar spiral MEAs were first made in a cleanroom using thin film fabrication processes similar to our previously reported work [24,25] (Figure 1A.1). First, 5 µm polyimide was spin-coated on silicon wafer. Next, 0.5 μm amorphous silicon carbide (a-SiC) was deposited on the polyimide at 350 °C using plasma-enhanced chemical vapor deposition (PlasmaTherm Uniaxis), with power of 270 W, pressure of 1000 mtorr, and 164 sccm Ar, 600 sccm SiH4/Ar, and 36 sccm CH4 gas flow rates. (Figure 1A.2). Titanium-gold-titanium layers (30, 300, 30 nm, respectively) were then deposited using plasma sputtering (AJA International ATC-2200 Sputter System). Photolithography processing and wet etching were used to define the metal traces with the width of 20 μm (Figure 1A.3). In this step, a thin layer of photoresist (Micro posit S1805, Kayaku Advanced Materials Inc., Marlborough, MA, USA) was used to pattern the metal traces and titanium and gold etchant (buffer oxide etch 1:10 and gold etchant TFA, respectively). A subsequent 1 µm layer of a-SiC was deposited, followed by a coating with a 3 μm polyimide layer to electrically insulate the conductive traces (Figure 1A.4). Photolithography was used to pattern the electrode site vias and bond pad areas and define the overall shape of the MEA (Figure 1A.5). In this step, the top polyimide was etched using O_2_ plasma (200 W power, 200 mtorr pressure, 25 sccm O_2_, and an a-SiC layer was etched using a mixture of O_2_ and fluorine-based plasma (SF6) (200 W power, 120 mtorr pressure, 2 sccm O_2_, 6 sccm SF6 gas flow) in a reactive ion etching system (RIE). After accessing the metal in the electrode vias, another photolithography pass was performed solely to pattern the electrode sites. In this lithography process, a bilayer of resists (lift-off resist (LOR5A, Kayaku Advanced Materials Inc., Marlborough, MA, USA) and photosensitive resist (SPR 220.7)) were coated and patterned to facilitate the lift-off process. Next, sputtered iridium oxide film (SIROF) was deposited in a plasma sputtering chamber, covering the entire wafer. The SIROF was then lifted-off in a resist stripper solution, leaving SIROF only on the 50-μm-diameter electrode sites (Figure 1A.6). Lastly, a relatively thick layer of photoresist (PR, 15 μm) was uniformly spin-coated over the entire wafer to tolerate the upcoming etching process. In this photolithography step, individual device outlines were patterned. Background bottom a-SiC and bottom PI were etched in a RIE chamber to create the individual layout of spiral array. First, a-SiC was etched with mixture of O_2_ and fluorine-based plasma (SF6) and then the bottom layer of polyimide was etched by changing the gas to pure O_2_ (Figure 1A.7). Residual PR was removed by a PR stripper. Lastly, in this stage, the wafer was washed with deionized water (DIW) and soaked overnight in 87 °C DIW for structure release. Polyimide acts as a release layer, allowing the array to be removed from the silicon carrier wafer when soaked in the DIW (Figure 1B). A silicone-based pillar with a height and diameter of 1000 and 100 μm, respectively, was printed on a cell culture glass slide using UV-cured silicone- base adhesive (Dymax 204-CTH-F-VT-MD) extruded from a 31G needle and UV cured. The center of spiral MEA was then manually centered and laid on the 1 mm pillar using sharp-tip forceps while the outer side of the MEA was stretched and glued to the cell culturing glass slide (adhesive epoxy Loctite EA-M121 HP), forming a 3D helical MEA from a planar spiral MEA (Figure 1C,D). Scanning electron microscopic (SEM) images of a representative 3D helical structure are shown in Figure 1E. On the distal end, a micro strip connector (Omnetic -Dual row SMT) was soldered to the gold bond pads and stainless reference wires. Finally, a cylinder for cell culturing, with 10 mm height and 10 mm outer diameter was placed and sealed around the helical MEA. Pictures of a culture-ready 3D helical MEA are shown in Figure 1D.

### 2.2. Electrochemical Testing

In order to demonstrate the functionality of 3D-Helical MEAs for neural recording, devices were subject to electrochemical impedance spectroscopy (EIS) and cyclic voltammetry (CV) using a Gamry potentiostat (Gamry Instrument 600+, Gamry Instruments, Warminster, PA, USA). EIS measurements were carried out over a range of 1 Hz–100 kHz for all 16 SIROF electrodes in four 3D-Helical MEAs using a sinusoidal voltage with an RMS amplitude of 10 mV. Figure 2A shows impedance of all 16 channels for a representative MEA and Figure 2B includes the measured magnitude of impedance at the frequency of 1 kHz for all four samples.

The CV data were acquired at two sweep rates of 50 mV/s and 50 V/s between −0.6 to +0.8 (water oxidation-reduction limit) versus Ag|AgCl in phosphate buffer saline (PBS, pH 7.4). Charge storage capacity was calculated from the cathodal area of time-current curve during a complete cycle. CSC was calculated to be about 50 mC/cm^2^ at the sweep rate of 50 mV/s which is similar to previously reported CSC for SIROF electrodes by Maeng et al. and Cogan et al. [26,27]. Figure 2C,D shows representative CVs of SIROF electrodes in 3D Helical MEAs for both sweep rate of 50 mV/s and 50 V/s.

### 2.3. D iPSC Culture and Pharmacology

Cryopreserved iPSC sensory neurons (SNs) were purchased from Anatomic, Inc (RealDRG Nociceptors, Fisher Scientific, Pittsburgh, PA, USA) and stored in liquid nitrogen, upon arrival, until the seeding date. On the day of seeding, SNs were thawed, centrifuged, aspirated, and resuspended in a nutrient rich medium consisting of thermally-sensitive polymerizable type-I collagen (Sigma Aldrich, Burlington, MA, USA) supplemented with Stem Cell Qualified ECM Gel [1:80] (Millipore Sigma), and Chrono-Senso-MM (Anatomic), for an overall collagen concentration of [3.46 mg/mL]. To create a 3D ECM taller than 1 mm with a diameter greater than 1.4 mm, a 10 µL droplet containing 140 k SNs was pipetted directly over the array and allowed to permeate the helical structure. Cultures were then immediately placed in an incubator and kept at 37 °C with 10% CO_2_ in 95% humidity for two hours to actuate collagen polymerization. Once the 10 µL seeding droplet was polymerized (illustrated in Figure 1F), the wells were flooded with 200 µL of Chrono-Senso-MM (Anatomic). Throughout the culture duration, cells were kept in Anatomic’s proprietary Chrono Senso-MM maintenance medium; with 50% medium exchanges occurring every alternate day. MEAs wells were topped with a soft Teflon lid and placed in 100 mm polystyrene petri dishes with additional 35 mm water reservoirs to maintain humidity.

### 2.4. MEA Recordings and Data Analysis

All recordings took place within the CO_2_ incubator, maintaining 37 °C, 10% CO_2_, and upwards of 95% humidity. Extracellular action potentials (EAPs) were recorded using a Multichannel Systems M2100 with 16-channel amplifier (ME2100-μPA16). Custom helical arrays were connected via Omnetics A79039-001 socket (NSD-18-DDGS). Continuous data was collected simultaneously across all channels at 20 kHz sampling rate and filtered in a parallel continuous recording using a 200–3500 Hz 4-pole bandpass Butterworth filter. EAPs were detected using Plexon’s Offline Sorter software (version 4, Plexon Inc., Dallas, TX, USA) by setting ± 4.5σ thresholds, calculated from the entire continuous recording from each respective channel. Reported ‘Spike counts’ were cumulative threshold crossings per channel from a 10-min recording from each treatment/condition. Single units were identified manually based on separation in 2D principal component space and averaged using Plexon’s Offline Sorter software.

### 2.5. Immunocytochemistry and Imaging

Immediately following 25 µM KCl exposure, cells were washed once with sensory maintenance medium and then fixed by incubating cultures with ice-cold 4% paraformaldehyde (PFA) for 30 min. PFA was then removed, and cultures washed twice with ice-cold 1x phosphate buffered saline (PBS) and stored at 4 °C until time of antibody labeling. Cultures were permeabilized using 0.5% Triton-X100 and blocked for 3 h with 4% normal goat serum. Cultures were then incubated with 0.5% concentration rabbit species anti-NF200 (Abcam, Cambridge, UK) overnight at 4 °C, washed three times with 1 × PBS, and subsequently incubated for 4 h with 0.4% concentration Goat anti-rabbit Alexa Fluor 555 nm wavelength (Abcam vendor) in combination with 0.06% DAPI. Cultures were washed 3 times for at least 30 min with 1x PBS, wrapped in parafilm, and stored at 4 °C until imaging. Confocal images, including z-stacks, were collected using a 10 × objective and the Leica SM8 laser scanning confocal microscope (Leica Microsystems, Wetzlar, Germany). Scanning parameters were set, and images stored, using Leica LAS X control software (version 5.1). Maximum intensity projections and 3D interpolation images were generated using ImageJ software (version 1.53q, NIH, USA).

## 3. Results

### 3.1. Preliminary Electrochemistry

To determine if our 3D helical MEAs were capable of recording EAPs, we performed electrochemical measurements (EIS and CV). Figure 2A shows mean frequency-dependent impedance and associated phase values for all 16 electrodes from a single MEA and demonstrates electrical continuity across the device. Figure 2B shows the average impedance magnitude at 1 kHz for all electrodes across 4 fully fabricated MEAs. Recorded impedance magnitudes ranged from 13 to 60 kΩ, which is well within the range necessary for single-unit EAP recordings [28,29,30,31]. CV measurements, likewise, suggested charge storage capacities of 50 mC/cm^2^ at the sweep rate of 50 mV/s and redox peaks at 0.1 and 0.2 V versus Ag|AgCl (Figure 2C,D) consistent with SIROF coated microelectrodes [26,27] and well within ranges necessary for electrical stimulation of EAPs in vitro.

### 3.2. Extracellular Action Potential Recordings

To confirm that our 3D helical arrays were capable of recording from a 3D culture model, we seeded 4 helical MEAs with 140 k iPSC sensory neurons embedded in a thermo-polymerizing gel and recorded spontaneous or evoked activity for up to 28 days. Figure 3 shows viable soma and axonal projections throughout the seeded volume. The figure shows a filtered continuous trace of voltage signals from a single representative electrode. EAPs were readily observed to exceed the threshold set for EAP detection with an average SNR of 2.80 ± 0.45 across 16 active electrodes (≥1 spk/min) from a total of 3 recorded wells. Additionally, collections of characteristic waveform shapes (i.e., single units) could be manually sorted (Figure 4B) based on separation in 2D principal component space projections.

Due to relatively low EAP amplitudes, we ensured that threshold crossings were biologically sourced by treating cultures with either 50 µM tetrodotoxin (TTX), a competitive voltage-gated sodium channel antagonist, or 25 µM potassium chloride (KCl). Figure 4C shows representative raster plots of activity prior to and immediately following addition of these two compounds. As expected in neuronal cultures, TTX significantly reduced EAP firing (21.8 ± 2.1 versus 6.2 ± 1.6 spikes, *p* = 0.01 Mann-Whitney), while KCl significantly, but transiently, increased EAP firing (13.5 ± 3.3 versus 25.8 ± 7.0 spikes, *p* = 0.002 Mann–Whitney). In total, these results (Figure 4D,E) demonstrate that we were able to record biologically sourced EAPs from iPSC neurons using our novel 3D helical arrays.

## 4. Discussion

### 4.1. Comparison with Other In Vitro 3D Electrophysiological Recording Approaches

A number of publications demonstrate electrode arrays that are 3D, in that they are not parallel to, or flush with, the culture substrate. However, most do not extend recording sites beyond one-to-two cell thicknesses from the substrate. Recent approaches for 3D MEA fabrication have been reviewed elsewhere [13,32], and apply either (i) electrodeposition/electroplating to create carbon or titania nanotubes [33,34], fractal appendages [31,35], and mushroom-shaped electrodes [36], or (ii) electrically, machined, or chemically etched features on the substrate [36]. Exceptions to these two categories include a multi-layered, or stacked, approach presented in [37] as well as the aforementioned mechanical actuation approach presented in [20]. In short, few existing designs allow for true 3D (i.e., volumetric) recording from throughout an in vitro organoid or tissue, and it becomes challenging to compare across substantially different platforms due to various cells, culture conditions, conductive/insulating materials used as well as the various measurements reported. Still, it is worth noting that our impedance and CV measurements are within the range of those observed for other 3D MEAs. Table 1 directly compares the material and electrochemical features of our array with values from previously published designs. Regarding functional recordings, we have recorded EAPs from electrodes across our array: from 40 µm to 1000 µm above the culture substrate encompassing a volume of approximately 10 mm^3^.

### 4.2. Limitations and Potential Challenges

The MEA presented here is, of course, not without its limitations and potential challenges. For example, reliance on manual, hand-placement of the spiral MEA on the central post presents two potential problems. One, this is a potential catastrophic failure point in device assembly as well as during cell seeding, as the spiral leads and the small adhesion point at the top of the post are fragile. Even if the leads are not damaged, it would still be catastrophic for the spiral to detach; rendering the MEA effectively 2D. The latter happened to one fully-fabricated MEA during cleaning and preparation, reducing our yield from 5 to 4. To answer this challenge, we can imagine a fitted post and spiral mask set that enables precise placement of the MEA over the post such that no subsequent manual placement step is required. The second potential problem associated with the post is related to imaging. The cultured tissue directly above the post, at the uppermost two electrodes, will be obscured. Future iterations of the device may use a thinner post, but this is an inherent limitation of our design. Lastly, there is the question of electrode yield. It is important to note that there is no inherent restriction on our design’s electrode density. However, there are practical limitations. Increasing the electrode number will result in increased spiral arm width and/or increased helix diameter. The latter option will result in increased ‘unrecorded’ volume beneath the helix. It is important, however, to note that 16 electrodes per well is an adequate number for meaningful pharmacology. In our previous work [38] using 2D electrode arrays, we demonstrated that a minimum of 4 active electrodes was sufficient for producing a drug screening assay based on primary dorsal root ganglion neurons.

### 4.3. Advantages of 3D In Vitro Culture/Recordings

There is a growing body of evidence to suggest that 3D versus 2D cultures of hiPSCs may better recapitulate both the native architecture and function of primary neural units (i.e., neurons and their non-neuronal support cells) [39,40]. Specifically, ref. [39] demonstrated that 3D neuronal cultures may differentiate heterogeneous populations consistent with anatomical organization and map transcriptionally to in vivo fetal development. Most importantly, 3D cultures have exhibited enhanced coordinated spontaneous activity and functional synapses in vitro, suggesting greater rates of phenotypic maturation [41]. To our knowledge, and to date, there has only been one published study of hiPSC nociceptors cultured in 3D [42]. This work did not include any direct functional measures (e.g., electrophysiology or calcium imaging) of either the 2D or 3D cultures, relying on the measured secretion of either substance P or CGRP. Since a primary phenotypic indicator of mature nociceptors and nociception is the rate [43,44] and pattern [45,46,47] of action potential firing, it is imperative that, in fundamental hypothesis or drug discovery studies, action potentials be measurable in a culture platform that promotes maximum maturation.

## 5. Conclusions

Here, we have demonstrated a novel, helical 3D MEA for in vitro EAP recording. To the best of our knowledge, our original design is the first to enable true volumetric recording based on the intrinsic flexibility of patterned thin films. It consists of established, biocompatible MEA materials and is fabricated using standard clean room procedures. Importantly, our highly adaptable design does not intrinsically limit channel count, helix diameter, or height, allowing integration of various tissue sizes and types. Moreover, the separate fabrication of the central post and two-part assembly will enable future iterative designs and integration of microfluidic and/or optical components for multi-modal stimulation and recording.

## Figures and Tables

**Figure 1 micromachines-13-01692-f001:**
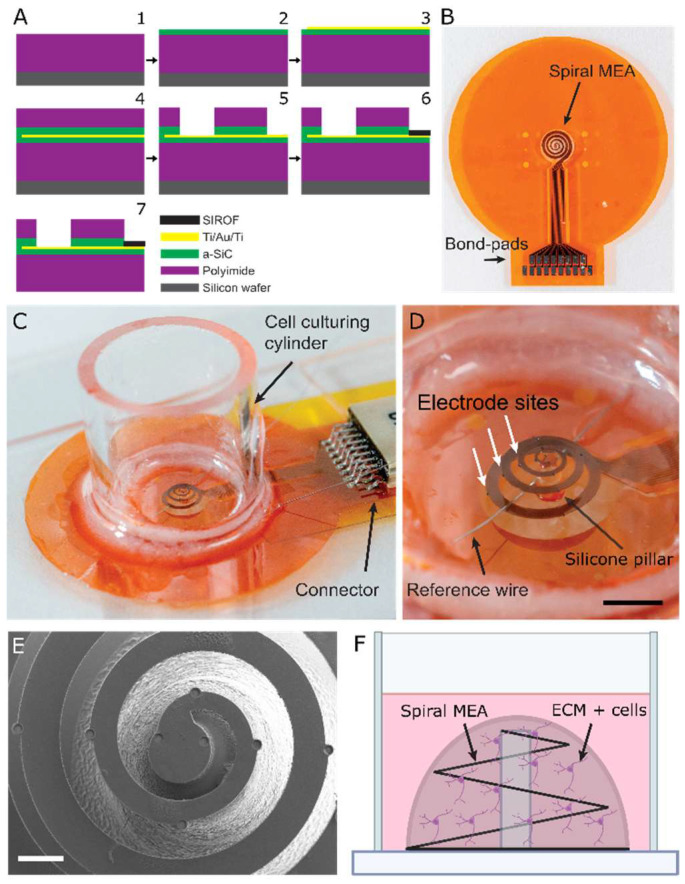
3D spiral electrode fabrication and assembly. (**A**) The layered fabrication process of 2D spiral MEA and (**B**) the resulting 2D spiral MEA. (**C**) Representative image of fully fabricated and 3D-assembled MEA and (**D**) a close-up view of the same. Electrode sites (example positions indicated by white arrow in **D**) appear as small black circles (50-µm diameter) along the length and height of the deployed spiral. Scale bar represents 2 mm. (**E**) Top-down scanning electron micrographs of fully-fabricated and assembled helical MEA. (**F**) Illustration of ECM (collagen and stem cell matrix) droplet residing on and around the spiral MEA, now extended as a helical MEA.

**Figure 2 micromachines-13-01692-f002:**
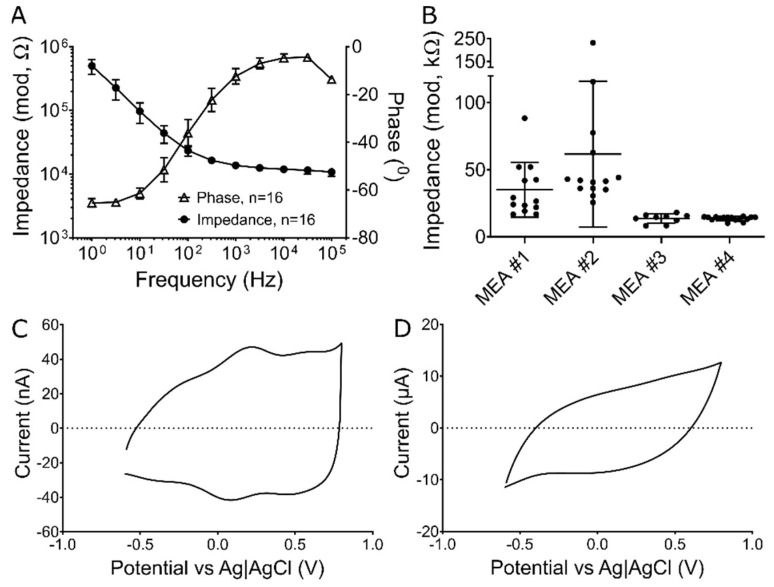
3D helical MEA electrochemistry. (**A**) Electrochemical impedance spectroscopy (EIS) for 16 SIROF coated electrode channels on a single selected MEA. Data points and error bars represent mean and standard deviation (SD). (**B**) Mean impedance magnitude and SD for SIROF-coated electrodes in four fully-fabricated MEAs at 1 kHz. (**C**) Representative cyclic voltammogram in PBS versus Ag|AgCl with sweep rate of 50 mV/s and (**D**) 50 V/s.

**Figure 3 micromachines-13-01692-f003:**
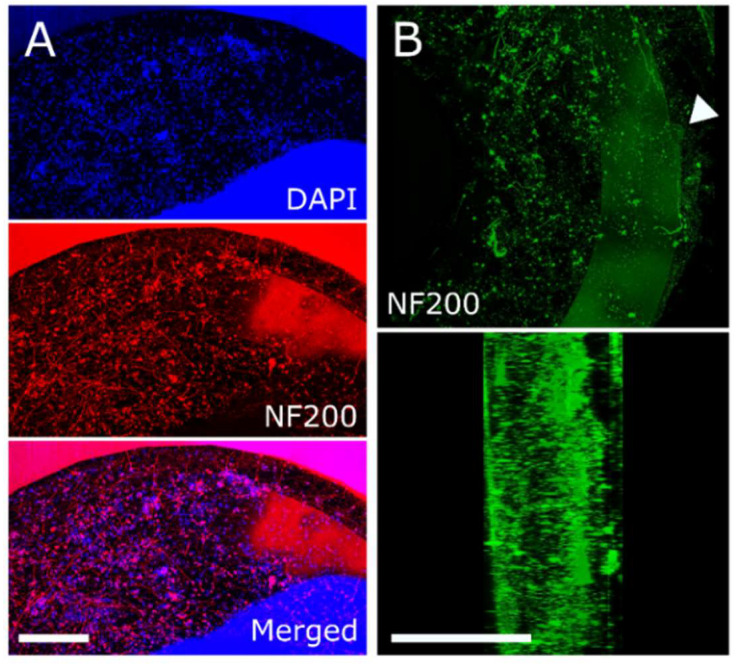
Viable soma and axonal projections present throughout at least 500 µm of recording volume following 28 DIV, indicated via immunocytochemistry. (**A**) Maximum intensity projection of DAPI (top, cell nuclei) and NF200 (middle, neurons) for the bottom-most 50 µm relative to the substrate. White arrowhead indicates location of recording electrode. Scale bar represents 100 µm. (**B**) Volumetric view of 500 µm section of neuronal culture (top, NF200, green). (Bottom) 90-degree rotated view of axial cross-section. Scale bar represents 500 µm.

**Figure 4 micromachines-13-01692-f004:**
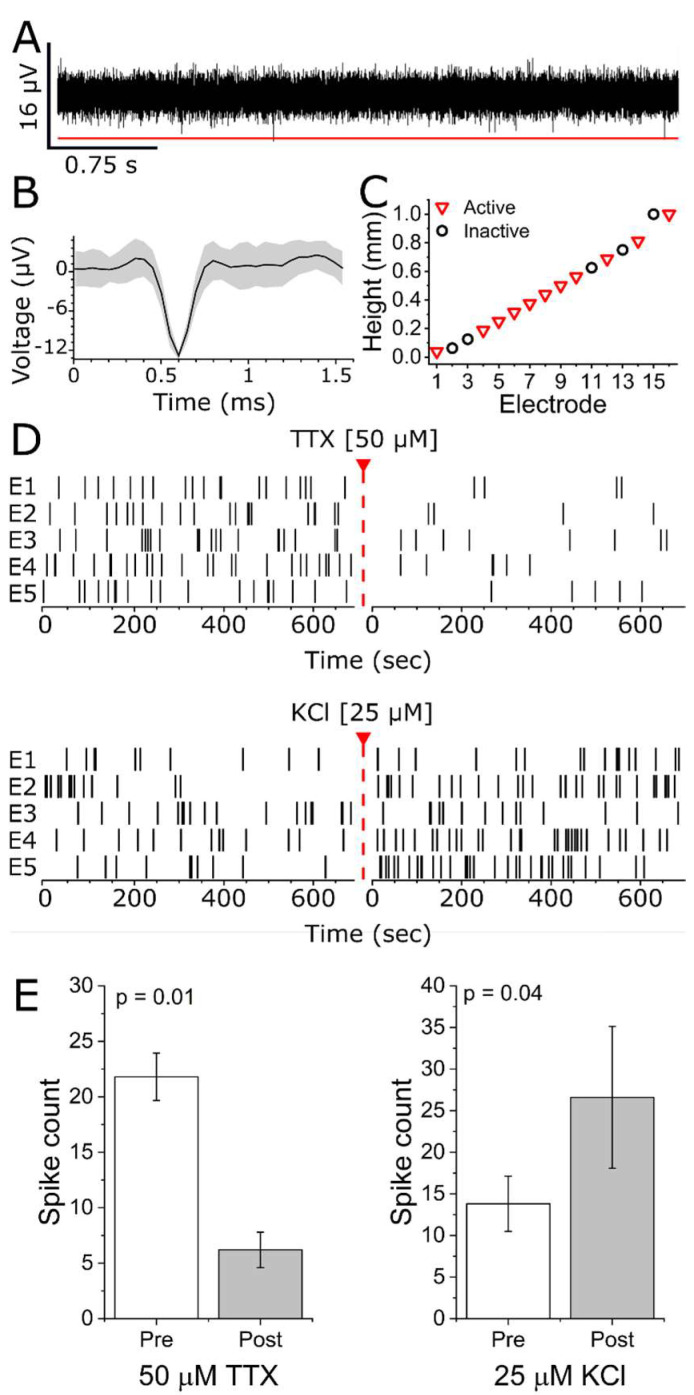
Spontaneous EAP recordings from iPSC sensory neurons on 3D helical MEAs. (**A**) Representative voltage versus time trace for a single electrode recorded at 20 kHz. Red line indicates 4.5σ threshold for detection of EAPs. (**B**) Representative traces of characteristic waveform collections (i.e., single units). The black line indicates the waveform average and the grey region the standard deviation. (**C**) Active electrode yield height distribution for a single helical array. Downward red triangle indicates electrode numbers/heights that recorded single units. Black circles indicate inactive electrodes. Heights are estimated based on pre-seeded configuration. (**D**) Representative raster plots from five electrodes on a single MEA exhibiting spontaneous activity (left) and 50 µM TTX-inhibited (top right) or 25 µM KCl-evoked (bottom right) recordings. The red triangle and dashed line indicate compound addition. Recordings were resumed as soon as possible following compound addition (<1 min). (**E**) Quantification and statistical analysis of TTX (left) and KCl (right) treatment spike counts. *p*-values were calculated using two-tailed Student’s *t*-test.

**Table 1 micromachines-13-01692-t001:** Direct material and electrochemical comparison between helical arrays (top row) and select published 3D MEAs.

Ref.	Electrode #	Size (µm), Material	Insulation	Impedance (kΩ)
	16	50, SIROF	Polyimide	13–60
[20]	80	50, Pt	Polyimide	37 ± 7 to 48 ± 6
[21]	16	20, PEDOT	SU-8	40–60 (estimate)
[17]	3–7	50, PEDOT:PSS	SU-8	3–80 (estimate)

SIROF: Sputtered iridium-oxide film; Pt: Platinum; PEDOT: poly(3,4-ethylenedioxythiophene); PEDOT:PSS: poly(3,4-ethylenedioxythiophene) polystyrene sulfonate.

## Data Availability

Processed MEA data and MEA CAD files are available upon written request.

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
