# Peer review of "A Novel 3D Helical Microelectrode Array for In Vitro Extracellular Action Potential Recording"

_micromachines, 2022, doi:10.3390/mi13101692_

Round 1

Reviewer 1 Report

This paper introduces helical thin film structure based concept for a 3D micro-electrode arrray to be used with 3D cell construct. The paper is mainly well-written and easy to understand and it contains sufficient technical and biological experiments to convince that the proposed concept works.

However, I believe that some more existing concepts could be easily found for the literature review. E.g. printed or carbonized needle electrodes were totally ignored. and there are also some references whose relevance can be questioned. In the methods section more details of the fabrication process should be given. Also information about electrode dimensions is missing and also the reader is hard to get image about the electrode and track layout as there is not any figure presenting them. Discussion section could be improved by adding more discussion about pros and cons of the proposed concept and the future visions.

More comments in the attached file.

Author Response

We thank the reviewer for their thoughtful suggestions and critiques. Please find a point-by-point response attached.

Reviewer 2 Report

After thoroughly reading the manuscript, I can say that it presents interesting information, with a certain degree of novelty. But in order to recommend its publication, it has to be fully reliable and consistent, this requiring some improvements.

Thus, I have some comments and recommendations:

- The English language used throughout the manuscript must be improved. Several grammar corrections and changes in terms of style are required for a better understanding of the meaning or/and of the scientific information.

- The quality of the figures must be improved.

- Section 2.2, line 117: please, specify the pH value of the phosphate buffer saline (PBS).

- Section 3.2: Figures 4B and 4E are not discussed in the text. Please, insert some comments related to these figures.

- It would be more useful to present together Section 3. Results and Section 4. Discussion. Besides, more discussions/explanations/correlations regarding the obtained results are required, in addition to those already presented by making a comparison with the data from the literature.

- Section 5. Conclusions do not sufficiently highlight the originality and the significant contributions of the study. Therefore, it must be completed.

- The references must be written consistently, in accordance with the requirements of the journal, comprising all the necessary details (the name of the journal, the names of all authors, the volume, the page number and the year). For example references 5, 9, 11, 13, 14, 20, 22, 24, 28, 29, 31, 33, 34 are not correct written / are incomplete.

In conclusion, the manuscript needs major revision before publication could be recommended.

Round 2

Reviewer 1 Report

Thank you for many improvements to your manuscript. However, still some concerns left:

1) Does the journal really require e-mails for each affiliation? Never seen that before.

2) Fig 1E and 1F not mentioned in the figure caption nor referred in the text. Please fix.

3) Motivation for using stainless steel(?) as the reference material instead of more commonly used Pt or Ag/AgCl?

4) Reference still missing for "...which is well within the range necessary for single-unit EAP recordings (i.e., < 1 MΩ)".

5) In fig 2B I would still use 150 as the max y value. Now just one untypical measurement point of MEA2 ruins the scaling for  MEA3 and MEA4.

6) "In our previous work [38] using 2D electrode arrays, we demonstrated that a minimum of 4 active electrodes was sufficient..." I don't think 38 is correct reference here. Please check it. And if something is sufficient in 2D, does that really apply to 3D? In my opinion no. If you disagree with me, please justify your opinion.

7) I still would like to see some figure which visualizes how the construct looks like when the cells are there. If you don't have any real photo, then please draw it.

8) One advice for the future publications, please answer to the reviewer comments one by one. Makes the reviewer much more happy if he or she does not have to guess which out of the ten concerns you have actually replied to.

Reviewer 2 Report

After carefully reading the revised manuscript and the responses given by the authors to my previous comments, I ascertained that the authors took into consideration some of my suggestions, made several modifications/corrections and completed the manuscript. Thus, I noticed an improvement of the manuscript, significant information being added.

In conclusion, my opinion is that the manuscript can be accepted for publication in present form.

Author Response

We thank the reviewer for their input and approval of the present form.

Round 3

Reviewer 1 Report

Thank you for your responses. I am happy now.